# Formation of the Traffic Flow Rate under the Influence of Traffic Flow Concentration in Time at Controlled Intersections in Tyumen, Russian Federation

**Viacheslav Morozov * and Sergei Iarkov**

Department of Road Transport Operation, Industrial University of Tyumen, 38 Volodarskogo Street, 625000 Tyumen, Russia; jarkovsa@tyuiu.ru
* Correspondence: morozov1990_72@mail.ru or morozovvv@tyuiu.ru; Tel.: +7-982-943-59-40

**Abstract:** Present experience shows that it is impossible to solve the problem of traffic congestion without intelligent transport systems. Traffic management in many cities uses the data of detectors installed at controlled intersections. Further, to assess the traffic situation, the data on the traffic flow rate and its concentration are compared. Latest scientific studies propose a transition from spatial to temporal concentration. Therefore, the purpose of this work is to establish the regularities of the influence of traffic flow concentration in time on traffic flow rate at controlled city intersections. The methodological basis of this study was a systemic approach. Theoretical and experimental studies were based on the existing provisions of system analysis, traffic flow theory, experiment planning, impulses, probabilities, and mathematical statistics. Experimental data were obtained and processed using modern equipment and software: Traficam video detectors, SPECTR traffic light controller, Traficam Data Tool, SPECTR 2.0, AutoCad 2017, and STATISTICA 10. In the course of this study, the authors analyzed the dynamics of changes in the level of motorization, the structure of the motor vehicle fleet, and the dynamics of changes in the number of controlled intersections. As a result of theoretical studies, a hypothesis was put forward that the investigated process is described by a two-factor quadratic multiplicative model. Experimental studies determined the parameters of the developed model depending on the directions of traffic flow, and confirmed its adequacy according to Fisher's criterion with a probability of at least 0.9. The results obtained can be used to control traffic flows at controlled city intersections.

**Keywords:** traffic congestion; traffic flows; traffic organization; traffic management; controlled intersections; intelligent transport systems

## 1. Introduction

One of the unsolved problems for city transport systems to date is the problem of increasing the efficiency of traffic management in terms of preventing traffic congestions [1]. Their formation in the road network inevitably entails a number of negative consequences, the most tangible of which for the urban population is an increase in the time of movement within the city due to an increase in transport delays [1], excessive fuel consumption by cars [1], environmental deterioration [1,2], and a decrease in the level of social comfort and quality of life [1,3]. In this regard, this problem is relevant and represents a serious challenge for most of the administrations of large cities, engineering, and science.

In many previous works, the main reason for the formation of traffic congestion is said to be the combination of a high level of motorization and a lag in the development of the road network. [1,4,5]. In other words, there is a significant numerical difference between transport demand and transport supply, which is quantitatively comparable in a ratio of 4 to 1. It is obvious that when vehicles move across the city, various sections of the road will conditional locations where transport demand is either formed or satisfied [1,6]. Consequently, the formation of traffic congestion is a consequence of a decrease in the

traffic flow rate on the section of the road network serving traffic flows in relation to the section of the road network that forms traffic flows [7]. Thus, traffic flow rate should be taken as a target indicator of the study.

Currently, in science and world practice, there is no one way to solve the problem of increasing the efficiency of traffic management in cities in terms of preventing traffic congestion on the road network. Conventionally, three global approaches can be distinguished: the road-building approach [1,8], the organizational and administrative approach [1,8–19], and the approach consisting of the use of intelligent transport systems [8,20–24]. The names of the approaches reflect a set of key measures that are proposed to resolve transport problems. The road-building approach consists of improving the existing and designing a new road network and its infrastructure facilities [1,8]. The essence of the organizational and administrative approach lies in all kinds of restrictions on the movement of cars and the development of public transport, including alternative ways of moving around the city [1,8–19]. The third approach, mentioned above, involves introducing intelligent transport systems, as well as their subsystems at various levels [8,20–24]. It should be noted that today, each of these approaches is used both individually and in combination, and can be effective depending on the goals and objectives of the researcher, engineer, or city manager, as well as the available resources.

Current expertise shows that the creation of an effective traffic-management system is impossible without the use of modern intelligent technologies. Traffic flow management on the road network in many cities and towns, as well as metropolitan cities in the USA, Japan, many European countries, Russia, and other developed countries of the world, is carried out by means of automated traffic control systems based on constantly updated data on the traffic flow rate.

Information about the traffic flow rates on sections of the road network is obtained by means of vehicle identification detectors [25]. However, for a complete understanding of the traffic situation on the investigated section of the transport network, it is impossible to restrict ourselves only to data on the flow rate. For an objective assessment of the state of the traffic flow, it is necessary to compare the actual number of moving vehicles with the measure of the flow concentration either in space or in time [25,26].

Until the end of the last century, traffic flow theory used a spatial measure of concentration—traffic flow density [26]. However, modern scientific research proposes to switch to the use of traffic flow concentration in time—lane occupancy. According to a number of researchers, the process of measuring the temporal concentration of traffic flows requires much lower economic and labor costs, and the data obtained are more valid [25–33].

Regardless of the city and even the country where the processes aimed at increasing the efficiency of traffic management are implemented, the priority for any state is to ensure safety, preserve the life, and maintain the health of citizens [1,8,25]. Therefore, it is not surprising that against the background of the growth of motorization in cities, there is an increase in the number of traffic lights on the road network. The Russian Federation is no exception because, according to Russian legislation, a large-scale installation of traffic-light control devices is carried out on the sections of the road network with high traffic congestion and increased accident rates. The current regulatory and technical documents clearly stipulate the rules and conditions for the use of traffic lights, which in three out of four cases are directly related to the traffic flow rate. The purpose of a traffic light as a technical means of organizing traffic is to increase the level of road safety. The presence of traffic light control means is one of the significant factors limiting the maximum possible value of the traffic flow rate, which directly depends on the ratio of traffic light signal durations in the control cycle [34–37].

Therefore, the purpose of this work is to establish the regularities of the influence of the concentration of traffic flow in time on the traffic flow rate at controlled city intersections.

This paper presents an analysis of the state of the issue of the indicated problem using the example of Tyumen, Russian Federation, based on which the purpose of the study was formulated.

The methodological basis of the research necessary to achieve this purpose is presented. The structuring and definition of the boundaries of the system under study are shown. The results of selecting the most significant factors influencing the traffic flow rate are presented.

The results of modeling the investigated process are given, which made it possible to formulate a working hypothesis about the type of model that reflects the process of changing the traffic flow rate under the influence of the selected factors.

The results of experimental studies carried out in order to confirm or refute the working hypothesis are presented.

## 2. Analysis of the State of the Issue

The results of the analysis of statistical data [38] showed that over the past 20 years, the level of motorization in Tyumen has increased almost threefold, and according to various estimates, is now approaching the mark of 590 cars/1000 people (Figure 1). The study of the structure of the motor vehicle fleet showed that more than 80% of all cars in the city belong to the category of light passenger vehicles (Figure 2), which, according to the authors' assumption, are more likely to be categorized as private vehicles.

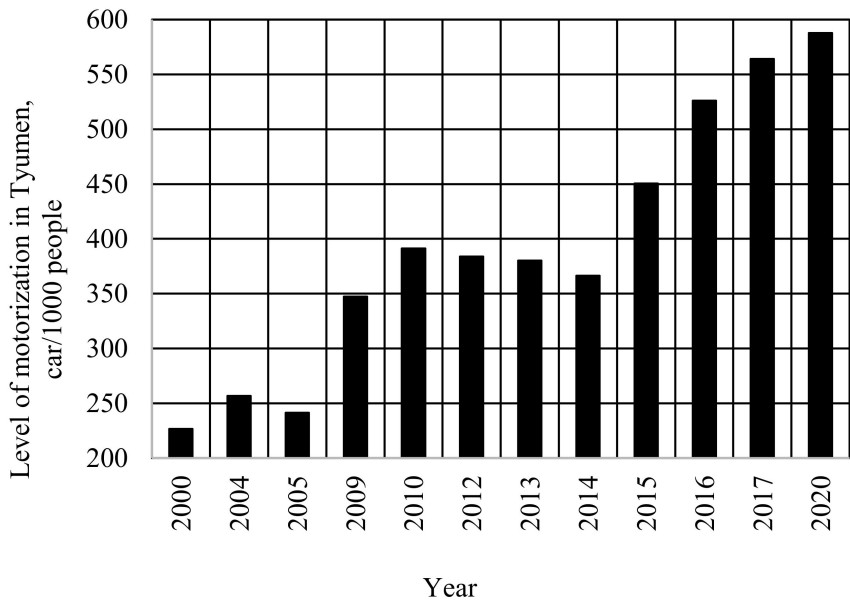

**Figure 1.** The dynamics of changes in the level of motorization in Tyumen [38].

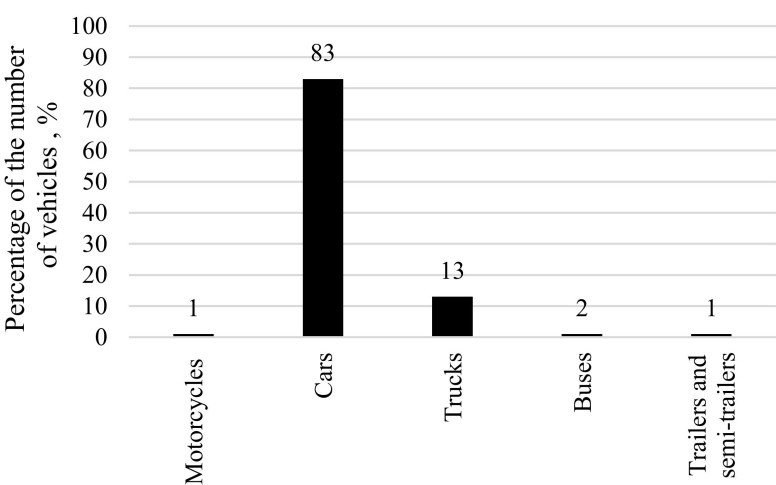

**Figure 2.** The distribution of the number of vehicles in Tyumen by category [38].

In his work [39], Goltz G.A. predicted that when the level of motorization in cities reaches 380 cars/1000 people, a spiral of automobile dependency is formed, which is a cyclical reproduction of the problems of the transport system of large cities at a higher level than the previous cycle. If the level of motorization reaches the threshold value of 500 cars/1000 people (i.e., on average, every two residents of the city have a car), the capacity of the road network is considered exhausted. With that, congestion is formed throughout the city's transport network.

Thus, the results of the analysis of the state of the current situation have shown that the problem of traffic congestion is also urgent for the city of Tyumen.

Taking into account the continued growth rates of motorization, traffic-light control devices are required in almost all key sections of the road network in the cities of the Russian Federation. Figure 3 shows a graph of changes in the number of traffic light units for the last nine calendar years in Tyumen [40]. The number of intersections equipped with traffic lights has also almost doubled over the past two decades, and has approached 400. The total number of traffic lights does not decrease, and is steadily increasing by an average of 11 units per calendar year. Placing traffic lights on the road network is one of the fastest and most effective ways to improve road safety. However, traffic-light control is also a factor in reducing the capacity of the intersection, which additionally limits the maximum value of traffic flow rate at city intersections [34–37] and exacerbates the situation.

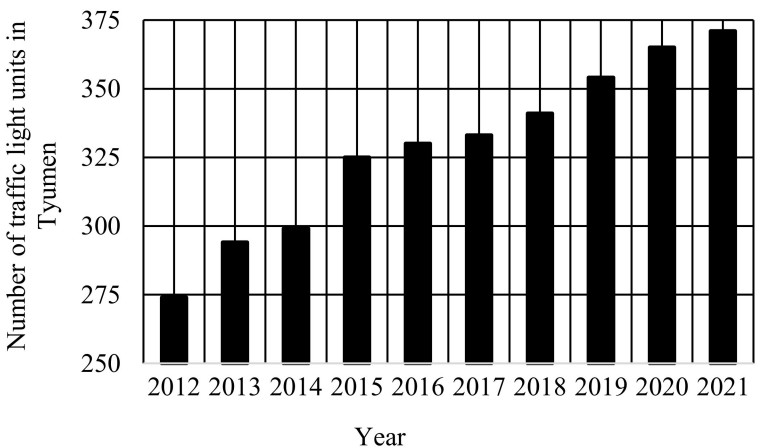

**Figure 3.** The dynamics of changes in the number of traffic lights in Tyumen [40].

The formation of a traffic jam can be considered both a stochastic and a completely deterministic process caused, on the one hand, by a random value of transport demand, weather and climatic conditions, emergency situations, and other factors; and on the other, by the geometric features of the road network, the presence of traffic lights on its sections, the road surface quality, and other quite predictable phenomena [41]. Therefore, in cases where vehicle detectors record a decrease in traffic flow rate, the following uncertainty arises: this situation can be caused by both a real formation of a traffic jam due to a traffic accident, vehicle breakdown, high load of the considered section of the road network, etc., and an actual absence of vehicles in the studied lane. In the fundamental theory of traffic flows, this uncertainty is resolved by comparing the data obtained on the traffic flow rate and its concentration [26].

The most common measure for assessing the concentration of traffic is its density $p$. Density $p$ means the number of vehicles per unit length of a section of the road network [25,26]. Until the second half of the 20th century, most of the mathematical models and studies were based solely on the use of the traffic-density indicator [41]. However, with the widespread use of vehicle detectors, which are one of the main elements of automated traffic control systems as part of intelligent transport systems, lane occupancy is proposed as the main measure of traffic concentration. Lane occupancy $\theta$ is the fraction of the total

duration of the measurement during which vehicles were in the control area of the detector. Lane occupancy $\theta$ is determined as follows [25,26]:

$$\theta = \frac{\sum_{i=1}^{n} t_i}{T}, \tag{1}$$

$$\theta = \frac{\sum_{i=1}^{n}(L_i + d)/u_i}{T}, \tag{2}$$

where $\theta$ is lane occupancy; $t_i$ is the time spent by the $i$-th vehicle in the control zone of the detector, s; $L_i$ is the length of the $i$-th vehicle passing through the control zone of the detector, m; $d$ is the length of the detector frame, m; $u_i$ is the speed of the $i$-th vehicle in the flow; and $T$ is the duration of the measurement, s.

The need to switch from using the traffic density indicator to using the lane occupancy indicator and the advantages of this transition are due to a number of arguments (Table 1).

**Table 1.** The comparison of indicators of traffic flow concentration [7,27–33].

| Comparison Criterion | Traffic Flow Density | Lane Occupancy |
|---|---|---|
| Physical meaning | Indicator of flow concentration in space | Indicator of flow concentration in time |
| Consideration of the influence of the traffic flow composition | Ignores the influence of vehicle length and flow composition | Considers the actual effect of the length of each vehicle, and therefore the composition of the traffic flow as a whole |
| Need to consider the speed characteristics of the traffic flow | Calculated using traffic-flow speed data | Includes traffic-flow speed characteristics |
| Point measurement | Not measured in case of point measurement | Possible to measure at a point of the road |
| Labor intensity of measurement, necessary equipment for measurement | Objective data can be obtained only through the use of detectors with two inductive loops, the installation, maintenance, and repair of which is associated with complex road construction works; the service life of such a system is one year | Measurements are taken using a single video detector with longer lifespan and mobility |

First of all, the feasibility of using the lane-occupancy indicator lies in its physical meaning. Both measures of traffic concentration are specific, but unlike density, occupancy is a temporal indicator. Of course, the length and area of roads and streets in cities are limited. The territory that could be adapted for the development of the road network is at a significant deficit. However, in case of emergency, it seems possible to delimit traffic flows in space through the construction of underground and aboveground transport infrastructure facilities. Time, however, is an inherently indivisible resource; its reserve is for all road users. In this case, as can be seen from Formula (2), lane occupancy also takes into account the impact of the length of each vehicle. In addition, for accurate measurement of traffic density, a more complex system is required, consisting of two control areas of the detector, based on the principle of operation of an inductive loop. Currently, this system is virtually not used for a number of reasons; more practical video detectors are used instead. Video detectors determine density by calculations performed on the basis of data on the flow rate and speed, which, in the event of a traffic jam, also creates uncertainty [7,25–33].

In this regard, it becomes necessary to use lane occupancy as the main measure of the traffic flow concentration. However, the results of the analysis did not reveal an accurate and unambiguous pattern of the influence of this indicator on the traffic flow rate, which determined the need for research in this direction.

## 3. Materials and Methods

### 3.1. Research Area

The city of Tyumen, a socially and economically developed regional center with a population of more than 815,000 people [38], located in Western Siberia, was chosen as the object for the study. As a transport hub, Tyumen is an important link and transport corridor for traffic flows not only from east to west, but also from north to south (and in the opposite directions). These routes intersect exactly in the central business district of the city. Despite the fact that trucks are prohibited in this part of the city, and dedicated lanes are allocated for urban passenger public transport, as in many other cities in different countries of the world, the central areas of the city experience excessive traffic congestion. This is especially noticeable in the morning, afternoon, and evening rush hours on weekdays. Therefore, a controlled intersection of Respubliki and M. Toreza streets was chosen as the investigated section of the road network. It is the center of the intersection of the transport routes from north to south, from west to east, and in the opposite directions.

### 3.2. Methodological Basis

The methodological basis of this study is a systemic approach that considers the objects under study as systems. More specific methods, on the basis of which this research was performed, are the existing, proven, and tested provisions of system analysis, traffic flow theory, experiment planning, impulses, probabilities, and mathematical statistics.

### 3.3. Preliminary Selection of Factors and Structuring of the System under Study

In order to implement a systemic approach as the methodological basis of this study, we decided to further study the process under consideration at the system level. To establish the boundaries of the system, we needed to determine the factors that are most significant in terms of the degree of influence on the target indicator of management—traffic flow rate. For this, based on previously performed studies, a complete list of factors was compiled, and a preliminary selection of the most significant of them was made. The factors influencing the change in the traffic flow rate were consolidated into the following groups [1,2,4,6,7,9,17,25–37,41–44]:

- Environmental conditions;
- Traffic flow state;
- Traffic conditions.

Table 2 presents a list of the main factors affecting the change in the traffic flow rate in cities, as well as their main characteristics.

**Table 2.** The results of preliminary selection of factors, their main characteristics.

| Group of Factors | Factor | Main Characteristics of Factors |
|---|---|---|
| Environmental conditions | Road surface condition Weather conditions | Dry or wet; icy Rain, snow, fog |
| Traffic flow state | Traffic concentration | Flow density, lane occupancy |
| | Flow speed characteristics | Average spatial speed, average speed over time |
| Traffic conditions | Traffic light control Geometric road characteristics | Traffic signal timing Number of lanes and their width; road surface quality |

In order to implement a systemic approach, the system under study was structured, the already established and scientifically confirmed relationships were identified, and the assumed connections between the elements of the systems were made. Figure 4 shows the designated elements of the system and the assumed connections between them in an enlarged form.

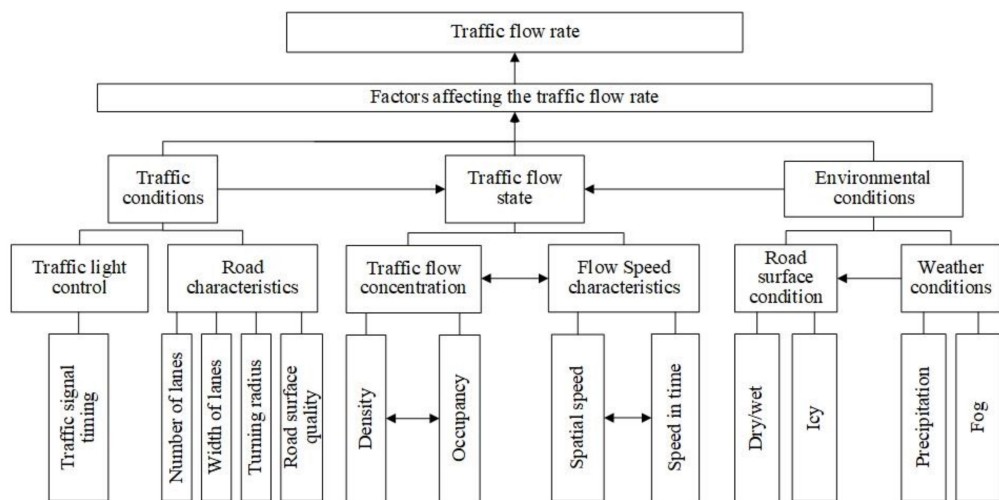

**Figure 4.** The results of structuring the system under study.

### 3.4. Establishing the Boundaries of the System under Study, Selecting the Main Factors

3.4.1. Regularities of the Influence of Environmental Factors

Key transport nodes (crossroads, intersections, junctions, etc.) located in the central part of cities, as well as other centers of attraction, considered the most problematic and of the greatest interest for further research, are part of the route network of urban passenger public transport. In accordance with the current legislative framework and national standards of the Russian Federation, measures are taken on these sections of the road network to ensure the safe operation of public transport, which implies more efficient snow removal, treatment of the road surface with reagents and other anti-icing agents, and other additional measures that change the influence of environmental factors by an undefined value. Therefore, the authors believe it is impossible to objectively assess the influence of environmental conditions on the change in the characteristics of the transport flow in this part of the study. In this regard, this work introduced a restriction for weather conditions and the state of the road surface. Further research was carried out under the condition that vehicles were moving on a dry road surface, in the absence of ice, precipitation, and fog.

3.4.2. Regularities of the Influence of the Factors of the Traffic Flow State

On the issue of the relationship between the indicators of traffic flow concentration in space and time, a number of studies have formed hypotheses about a possible linear relationship between traffic density and lane occupancy. [27]. However, there is also an opinion that this relationship is not always experimentally confirmed in practice [25,26,30]. Additional analytical studies were carried out to confirm or deny the possibility of a relationship between traffic concentration indicators. Formulas (1) and (2) presented earlier in this work can be considered equivalent. Formula (2) reveals the physical meaning of the lane occupancy indicator, and Formula (1) gives a more detailed idea of the occupancy measurement process and displays the principle of the detector operation. $\sum_{i=1}^{n} t_i$ in (1) is the sum of pulse durations [25] recorded by the detector when vehicles pass through its control zone. Therefore, to determine the relationship between the indicators of the traffic flow concentration, the temporal and spatial structures of the pulse measurement process for a homogeneous, uniformly moving flow of cars were considered. Based on the fundamental provisions of the impulse theory and taking into account the indicated conditions, it is possible to represent Formula (1) in the form of a value inverse to the average duty cycle [45]:

$$\theta = \frac{\sum_{i=1}^{n} t_i}{T} = \frac{\bar{t}}{T} n = \frac{1}{\bar{S}}, \tag{3}$$

where $\bar{t}$ is the average duration of the recorded pulse, s; and $\bar{S}$ is the average duty cycle.

Meanwhile, when studying the spatial structure of this process, the movement of cars in an enlarged form can be described by the value of the dynamic envelope [25,26,37,44]:

$$L_d = L + d_a,\qquad(4)$$

where $L_d$ is the dynamic envelope, m/car; $L$ is the length of the vehicle, m; and $d_a$ is the safe distance between moving cars, m. If the specified conditions are met, the variables $L_d$, $L$, $d_a$ in this case will be constant values.

Processes (3) and (4) are schematically shown in Figure 5a,b, respectively.

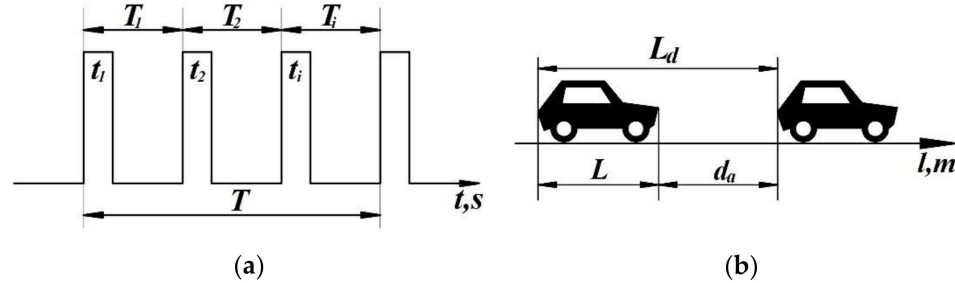

(**a**)  (**b**)

**Figure 5.** The temporal structure of the process of recording pulses (**a**) and the spatial structure of the process of car movement (**b**).

Comparing Figure 5a,b helped to expand the understanding of the presence of a possible relationship between the indicators of spatial and temporal traffic concentration. (Figure 6).

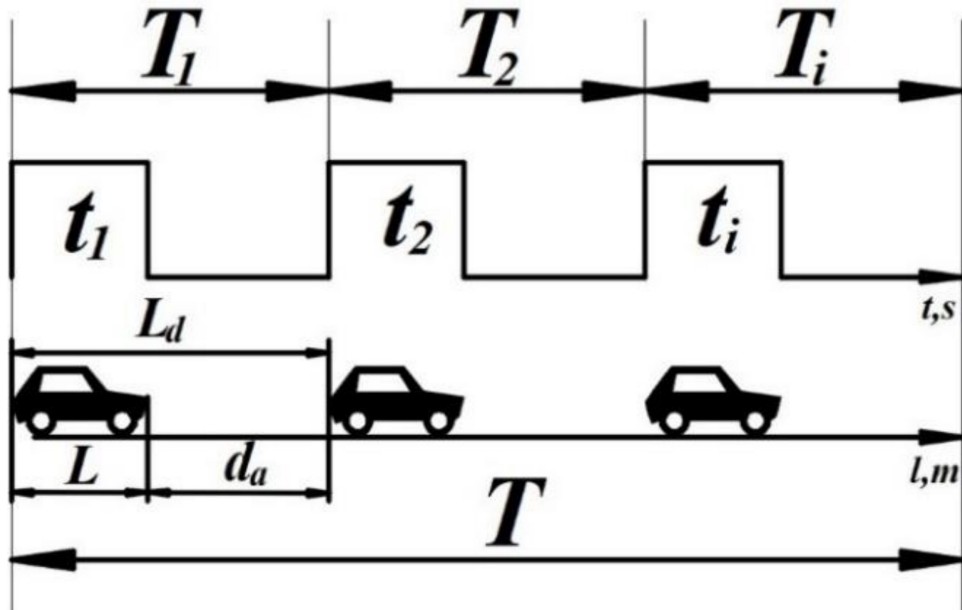

**Figure 6.** The spatial–temporal structure of the process of forming lane occupancy.

The results of studying the spatial-temporal structure of the process of forming lane occupancy make it possible to represent this process in the mathematical equation as follows:

$$\overline{S} = \frac{T}{n\overline{t}} = \frac{L + d_a}{L + d} = \frac{L_d}{L + d}.\qquad(5)$$

For $L_d$, the dimension is valid both in specific (m/car, km/car) and in absolute values (m, km). In the latter case, $L_d$ should be interpreted as the length of the part of the

lane in question containing the vehicle, including the distance provided by the driver for emergency braking. Therefore, having determined the considered section of the road network to the boundaries of the movement of one vehicle (from the rear bumper of the previous car to the rear bumper of the next one), it seems possible to express the dynamic envelope $L_d$ in terms of the traffic density $p$:

$$L_d = \frac{1}{p}. \tag{6}$$

Comparing (6) and (5) with (3), it becomes possible to establish the relationship between the occupancy of the lane and the density of the traffic flow:

$$\theta = (L + d)p. \tag{7}$$

Thus, the results of additional analytical studies have confirmed the validity of the assumptions about the presence of a linear relationship of traffic concentration indicators. In the case of extrapolating Formula (7) to the entire traffic flow and measuring $\theta$ in percent and $p$ in cars/km, Formula (7) is transformed as follows:

$$\theta = \frac{(L + d)}{1000} p \cdot 100\% = k_\theta p, \tag{8}$$

where $\theta$ is lane occupancy, %; $p$ is the traffic flow density, cars/km; and $k_\theta$ is the parameter of the relationship between the flow density and lane occupancy, $k_\theta = (L + d)/10$, km·%/car.

### 3.4.3. Regularities of the Influence of the Factors of Traffic Conditions

The mechanism of the influence of the traffic light control means on the traffic flow rate is carried out by the traffic light control cycle and consists of the following [25,26,34–37]:

$$Q_{ij} \leq c_{ij}, \tag{9}$$

where $Q_{ij}$ is the traffic flow rate at the exit of the controlled intersection in the $i$-th lane during the $j$-th phase of the traffic light cycle, cars/hour; and $c_{ij}$ is the capacity of the $i$-th lane during the $j$-th phase of the traffic light cycle, cars/hour.

In other words, the values of the traffic flow rate in each considered lane can increase or decrease under the influence of various factors, but cannot exceed the value of the lane capacity, the value of which is limited by the operation of traffic lights. In turn, the capacity of each lane at the exit of the controlled intersection depends on the specific characteristics of the traffic light control cycle [25,26,34–37]:

$$c_{ij} = M_{sij} \lambda_{ij}, \tag{10}$$

where $M_{sij}$ is the saturation flow in the $i$-th lane during the $j$-th phase of the traffic light cycle, cars/hour; and $\lambda_{ij}$ is the share of the effective duration of the $j$-th phase of the traffic light control cycle for the $i$-th lane.

Currently, the literature lacks an unambiguous formulation and a unified approach to determining the saturation flow. In a number of sources, the saturation flow is understood as the maximum possible traffic flow rate with the maximum saturated queue of the crossing point, which is partly comparable with the definition of capacity [25,26,34–37]. In some cases, we can assume these definitions as identical. Then, the saturation flow can be understood as the maximum possible capacity that can be achieved in the complete absence of the influence of the traffic light control cycle; i.e., $\lambda = 1$. Comparing Formulas (9) and (10), it is possible to determine the capacity of each considered traffic lane as the maximum possible value of the traffic flow rate, a part of which will inevitably be lost as a result of the influence of the traffic light control cycle.

Coefficient $\lambda$ is determined as follows [25,26,34–37,39]:

$$\lambda_{ij} = t_{ej}/T_c, \tag{11}$$

where $t_{ej}$ is the effective duration of the *j*-th phase of the traffic light control cycle, s; and $T_c$ is the duration of the traffic light control cycle, cars/hour.

The geometric parameters of the road (width and number of lanes) and the direction of the traffic flow (turning radius) affect the maximum value of $M_s$. Currently, there are various techniques for determining the value of saturation flow. In accordance with the current industry regulatory documents of the Russian Federation, the geometric parameters of the road and traffic direction are taken into account at the stage of designing controlled intersections and adjusting the existing modes of operation of traffic lights. Due to the fact that determining the saturation flow was not the goal of this study, to determine the value of $M_s$, a calculation method was adopted that also reflected the influence of the geometric characteristics of the road network.

Key controlled intersections of greatest interest for further research are part of the urban passenger public transport route network. In accordance with the current legislative framework of the Russian Federation, these sections of the road network require measures for the safe operation of public transport, which implies ensuring a high quality of the road surface (roadway). In this regard, the influence of the quality of the roadway was not considered in this work.

Ultimately, the results of the selection of factors showed that the most significant factors influencing the process of changing the traffic flow rate are: the concentration of the traffic flow in time, which is characterized by lane occupancy $\theta$, %; and the presence of traffic light control at the intersection, which is characterized by the influence of the share of permitting signal in the traffic light cycle $\lambda$ on the saturation flow $M_s$, cars/hour, depending on the geometric characteristics of the road and the direction of movement of the traffic flow in question.

### 3.5. Development of a Model of the Investigated Process

3.5.1. Development of a Mathematical Model of the Influence of Traffic Flow Concentration in Time on Traffic Flow Rate

To develop a model of the influence of lane occupancy on the traffic flow rate, the existing models of the traffic flow theory were analyzed [26,41,44]. For further research, we decided to develop a model based on the principle of macromodeling, considering the traffic flow as a whole. This approach can be used more effectively to solve problems in the field of traffic management [7,26,41]. In accordance with the basic provisions of the traffic flow theory, the fundamental macroscopic model describing the movement of a single-lane traffic flow is the Lighthill–Whitham–Richards hydrodynamic model [26,44]. The Lighthill–Whitham–Richards model provided significant clarity with minimal effort in the research conducted with its use [26,41]. This model was developed in the middle of the 20th century, but at the same time it has significant weight and value, and has a relationship with many modern models; for example, with the Prigogine–Herman synergetic model [46]. The main criticism of the Lighthill–Whitham–Richards model is its inoperability at low densities, which is reflected in the discrepancy between the theoretical distribution curve and real data on traffic flow. Based on Table 1, it makes sense to assume that the discrepancy between theoretical and experimental data can be justified primarily not by the lack of adequacy of the mathematical model, but by the impossibility of obtaining correct measurements of the traffic flow concentration in space, which is density. As stated earlier, data on traffic concentration in time, which is characterized by lane occupancy, are more valid. In this regard, this work provides no weighty justification for refusing to use the indicated mathematical model.

In the Lighthill–Whitham–Richards macroscopic hydrodynamic model, the traffic flow is likened to the flow of a liquid, namely water. The model itself is based on two key postulates:

$$u(t, x) = u(p(t, x)), \tag{12}$$

$$\begin{cases} u\prime(p) < 0, \\ Q(p) = pu(p). \end{cases} \tag{13}$$

where $u(t, x)$ is the speed of the vehicle at a time $t$ in the vicinity of a point on the road with a coordinate $x$, km/h; $p(t, x)$ is the traffic density, cars/km; $u(p)$ is the function of the dependence of the speed of the traffic flow on its density, km/h; and $Q(p)$ is the function of the dependence of the traffic flow rate on its density, cars/hour.

Function (12) is comparable with the earlier obtained Greenshields model [26,42], and is also known as the equation of state of the traffic flow, which is described by a linear model $y = ax + b$, if $a < 0$, $b > 0$. Taking into account the established relationship of indicators of traffic flow concentration in space and time in (8), Function (12) can be represented as follows:

$$u(\theta) = u_f(1 - \frac{k_\theta \theta}{k_\theta \theta_j}) = u_f - \frac{u_f}{\theta_j}\theta, \tag{14}$$

where $\theta$ and $\theta_j$ are, respectively, the actual and maximum lane occupancy at which a traffic jam occurs, %; and $u(\theta)$ is the function of the dependence of the traffic flow speed on lane occupancy, km/h.

The function of the dependence of the traffic flow rate on its density (13) is also known as the fundamental traffic flow diagram [26,44]. The resulting Equations (8) and (14) allow us to transform the fundamental diagram in (13):

$$Q(\theta) = \frac{\theta}{k_\theta}u(\theta) = \frac{\theta}{k_\theta}(u_f - \frac{u_f}{\theta_j}\theta) = \frac{u_f}{k_\theta}\theta - \frac{u_f}{k_\theta \theta_j}\theta^2, \tag{15}$$

where $Q(\theta)$ is the function of the dependence of the traffic flow rate on lane occupancy, cars/hour.

In Equation (15), the value of the free movement speed $u_f$ on the considered section of the road network is constant and is limited only by the road safety conditions. In accordance with the current traffic rules of the Russian Federation, the maximum permitted speed within the city limits is 60 km/h, considering there are no additional technical means to regulate the speed limit. The critical value of $\theta_j$ theoretically corresponds to $\theta_j \approx 100\%$. It is also assumed that the coefficient of interrelation of the concentration indicators $k_\theta$ under the same traffic conditions and the conduct of the study takes on a constant value. Consequently, $u_f$, $\theta_j$, and $k_\theta$ are constants, which ultimately allows us to make an assumption: the process of changing the traffic flow rate under the influence of lane occupancy is described by a one-factor quadratic model:

$$Q = b_1\theta - a_1\theta^2, \tag{16}$$

where $a_1$, $b_1$ are model parameters, cars/(hour·%).

### 3.5.2. Development of a Mathematical Model of the Influence of Traffic Light Control on Traffic Flow Rate

The influence of the controlled intersection on the traffic flow rate is carried out by switching the signals of traffic lights, as well as their duration as part of the cycle [4,25,26,34–37]. In the absence of traffic light control devices, as well as in situations where the effective duration of the $t_e$ phase remains unchanged and becomes equal to the duration of the entire traffic light cycle $T_c$, which is typical for traffic lights with a calling phase for pedestrian traffic, it is equipped on sections of the road network with high transport demand. In this case, the maximum value of the traffic flow rate with the complete long-term absence of pedestrians takes on a value equal to the value of the saturation flow $M_s$. The total duration

of the signals prohibiting traffic and the intermediate time step of the traffic light control cycle for the traffic flow is wasted time; in other words, the time during which there is no movement, and the flow rate at the exit of the controlled intersection equals zero.

Consequently, the maximum possible number of vehicles at the exit of the controlled intersection will depend on the effective phase duration $t_e$, s, and the total duration of the traffic light control cycle $T_c$, s. The results of earlier studies [34–37] assume $t_e \approx t_o$, where $t_o$ is the duration of the permitting signal of the main time step of the traffic light control cycle, s. This assumption is also accepted in this work. To determine $M_s$ in the study, a computational method was adopted, which also allows taking into account the influence of the geometric parameters of the considered section of the road network.

Thus, traffic light control on a section of the road network will additionally limit the maximum possible value of the traffic flow rate due to the ratio of the duration of the permitting signal to the total duration of the traffic light cycle. The process of changing the traffic flow rate at the exit of the controlled intersection is described by a one-factor linear model:

$$\begin{cases} Q = M_s\lambda, \\ \lambda = t_o/T_c. \end{cases} \tag{17}$$

where $M_s$ is the saturation flow, car/hour; $\lambda$ is the share of the permitting signal in the traffic light cycle; $t_o$ is the duration of the permitting signal of the main time step of the traffic light control cycle, s; and $T_c$ is the duration of the traffic light control cycle, s.

### 3.5.3. Development of a Two-Factor Mathematical Model of the Combined Influence of Traffic Flow Concentration in Time and Traffic Light Control on Traffic Flow Rate

After establishing the type of one-factor models that reflect the process of changing the traffic flow rate under the influence of key factors, it becomes possible to develop a multi-factor model by arranging. For this, it is necessary to predetermine the type of the proposed multi-factor model that can be presented in a multiplicative or additive form [47]. To decide on the type of model, the nature of the studied regularity was analyzed according to the following algorithm:

- The presence of extrema in the studied function;
- The behavior of the response function $Y$ if $X \to 0$ and $X \to \infty$;
- Points of the factor space necessarily belonging to the graphic display of the response function;
- The nature of the influence of factors on the response function.

As a result of the arrangement, a two-factor mathematical model was obtained:

$$Q = (b\theta - a\theta^2)M_s\lambda, \tag{18}$$

where $Q$ is the traffic flow rate, cars/hour; $\theta$ is the lane occupancy, %; $a,b$ are model parameters, 1/%; $M_s$ is the saturation flow, cars/hour; and $\lambda$ is the share of the permitting signal in the traffic light cycle.

Ultimately, a hypothesis was put forward: the process of changing the traffic flow rate under the influence of lane occupancy and the traffic light control cycle is described by a two-factor quadratic multiplicative model.

### 3.6. Experiment Planning

To confirm the developed hypothesis, determine the numerical values of the parameters and statistical characteristics of the mathematical model, as well as test it for adequacy, experimental studies were planned and carried out (Table 3).

**Table 3.** The experiment planning results: the main characteristics of the experiment and their descriptions.

| Experiment Characteristics | Description of Characteristics |
|---|---|
| Experiment type | Passive |
| Method | Continuous monitoring during the movement of traffic flows on the road network of Tyumen |
| Conditions | Real traffic conditions on the road network of Tyumen |
| Duration of experiment | Working days from 07:00 to 23:00, for calendar years 2014 to 2019 |
| Studied values: | |
| - target | Traffic flow rate on each lane, cars/hour |
| Factors: | |
| - traffic flow concentration in time | Lane occupancy, % |
| - traffic light operating mode | Duration of permitting signal, s Duration of traffic light control cycle, s |
| Additional restrictions on the conditions: | |
| - road surface condition; | Dry asphalt/asphalt concrete pavement, no ice |
| - precipitation; | No precipitation |
| - fog. | Clear visibility (no fog) |

Data on the traffic flow rate, the occupancy of lanes, and the mode of operation of traffic lights were collected at the intersection of Republiki and M. Toreza streets, Tyumen. Experimental data was obtained using the equipment shown in Figure 7.

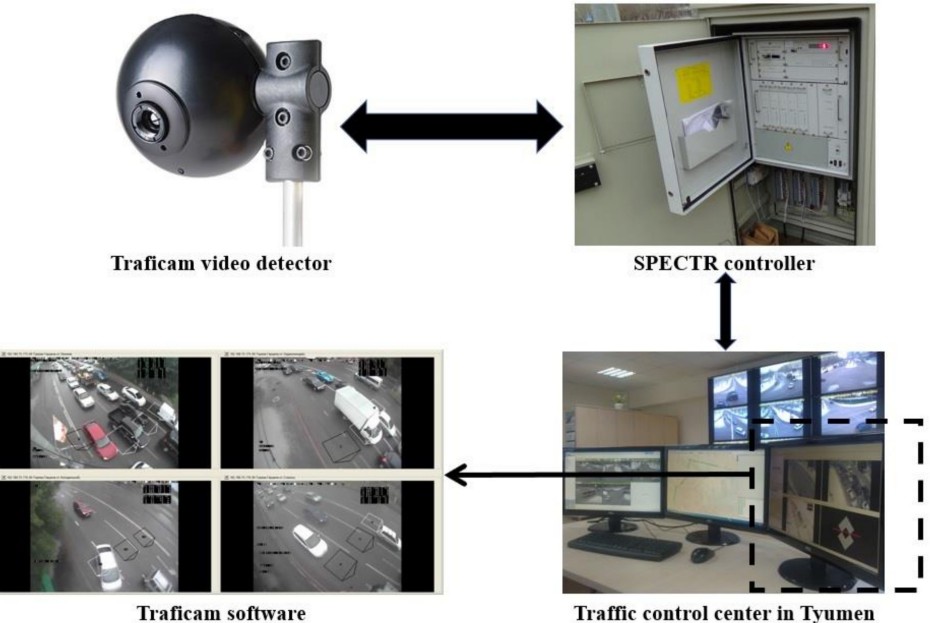

**Figure 7.** The interaction of equipment used to obtain the initial experimental data.

Experimental data on the traffic flow characteristics were obtained by means of video detectors installed at each approach to the investigated controlled intersection, with the control measurement zones in the area of the road marking of the stop line in each lane. The received data was transmitted via a fiber-optic line to the main data storage server of the traffic control center in Tyumen, connected to the automated workstation of the engineer of this center, which included a personal computer with specialized software.

Primary information about the mode of operation of traffic lights, including the geometric characteristics of the controlled intersection, the order of priority of passage and directions of movement of vehicles, and the location of the elements of the road traffic

light unit at the intersection of Respubliki and M. Toreza streets, was obtained based on documents provided by subordinate divisions of the administration of the city of Tyumen. Experimental data on the duration of traffic light signals as part of the regulation cycle were recorded through a traffic light controller installed at the intersection and then fed to the automated workplace of the engineer of the traffic control center in Tyumen.

The total sample of the initial data was 1184 measurements. In order to exclude the uncontrolled influence of the controlled intersection mode, the experimental data were preliminarily redistributed relative to the time of day corresponding to the operating time of a certain traffic light mode.

The obtained data were processed using the STATISTICA 10 software package. The statistical method of grouping by mean values was used as a grouping method.

Initial experimental data were obtained and processed using modern equipment and software: Traficam video detectors, SPECTR traffic light controller, Traficam Data Tool, SPECTR 2.0, and STATISTICA 10.

## 4. Results

### 4.1. Results of the Study of the Regularities of the Influence of Traffic Flow Concentration in Time on Traffic Flow Rate

For each operating mode of traffic lights, the experimental data were grouped according to the Sturges' formulas [48] for the number of intervals $k$ with a range $\delta$ of lane occupancy $\theta$, %. The grouping results are presented in Table 4.

**Table 4.** The result of grouping the experimental data.

| Traffic Light Operation Period | Traffic Direction | Number of Measurements | $k$, Number of Intervals | $\delta$, Interval Value ($\theta$, %) |
|---|---|---|---|---|
| 07:00–08:00 | Left-turn | 104 | 8 | 10 |
| | Forward | 104 | 8 | 11 |
| 08:00–09:00 | Left-turn | 72 | 7 | 12 |
| | Forward | 72 | 7 | 13 |
| 09:00–15:00, 19:00–23:00 | Left-turn | 720 | 10 | 9 |
| | Forward | 720 | 10 | 10 |
| 15:00–19:00 | Left-turn | 288 | 9 | 10 |
| | Forward | 288 | 9 | 10 |

Within the range, the type of distribution was determined [47], which, in most cases, corresponded to the normal distribution (Figure 8a). Using the least-square method [47], the regularity of the influence of lane occupancy on the traffic flow rate was established, the graphical representation of which is the regression line (Figure 8b).

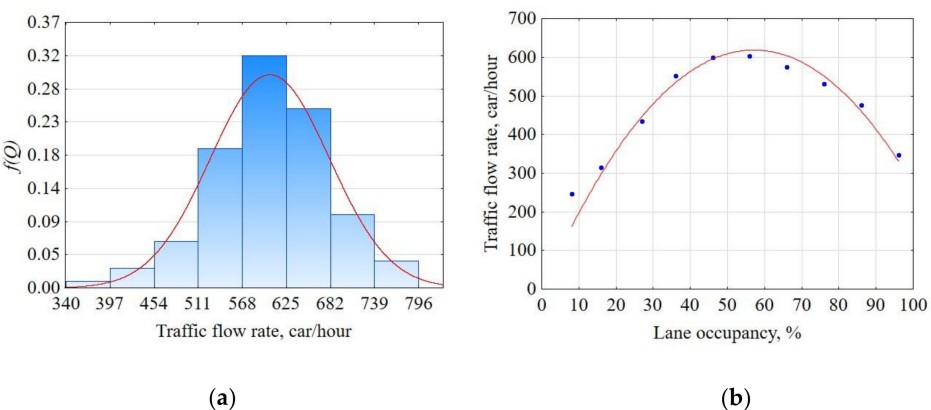

(**a**)  (**b**)

**Figure 8.** The histogram, the corresponding distribution function (**a**), and the regularity of the influence of the lane occupancy on the traffic flow rate (**b**) for the traffic lights' operating mode from 07:00 to 08:00.

For the mathematical model of the influence of lane occupancy on the traffic flow rate (16), the numerical values of its parameters were determined and are presented in Table 5.

**Table 5.** Numerical values of the parameters of the model of the influence of lane occupancy on the traffic flow rate.

| Traffic Light Operation Period | Traffic Direction | Model Parameters | | Optimal $\theta$, % | Maximal $Q$, Cars/Hour |
| --- | --- | --- | --- | --- | --- |
| | | $a_1$, Cars/(Hour %) | $b_1$, Cars/(Hour %) | | |
| 07:00–08:00 | Left-turn | **0.31** | 33.56 | 54 | 901 |
| | Forward | 0.09 | 12.36 | 73 | 450 |
| 08:00–09:00 | Left-turn | 0.25 | 30.35 | 61 | 923 |
| | Forward | 0.23 | 23.68 | 52 | 612 |
| 09:00–15:00, 19:00–23:00 | Left-turn | 0.29 | 28.48 | 49 | 695 |
| | Forward | 0.19 | 21.66 | 57 | 619 |
| 15:00–19:00 | Left-turn | 0.29 | 28.45 | 50 | 730 |
| | Forward | 0.27 | 26.43 | 49 | 649 |

In addition, for the model (16), statistical characteristics were determined and are presented in Table 6.

**Table 6.** Statistical characteristics of the model of the influence of lane occupancy on the traffic flow rate.

| Traffic Light Operation Period | Traffic Direction | Correlation Coefficient $r$ | Determination Coefficient $R^2$ | Student's $t$-Test | Fisher's Variance Ratio $F$ |
| --- | --- | --- | --- | --- | --- |
| 07:00–08:00 | Left-turn | 0.98 | 0.96 | 8.35 | 3.49 |
| | Forward | 0.95 | 0.9 | 7.44 | 2.90 |
| 08:00–09:00 | Left-turn | 0.98 | 0.96 | 10.86 | 3.27 |
| | Forward | 0.98 | 0.95 | 10.03 | 3.20 |
| 09:00–15:00, 19:00–23:00 | Left-turn | 0.98 | 0.96 | 8.36 | 3.13 |
| | Forward | 0.97 | 0.93 | 10.55 | 2.99 |
| 15:00–19:00 | Left-turn | 0.95 | 0.9 | 8.36 | 2.94 |
| | Forward | 0.96 | 0.91 | 8.64 | 3.05 |

The numerical values of the determination coefficients were in the range of 0.9 to 0.96, and the correlation coefficients were from 0.95 to 0.98, which indicated the presence of a very high-strength relationship between the variables. The excess of the calculated value of the Student's $t$-test compared to the tabular one confirmed the significance of the obtained correlation, and the Fisher's variance ratio $F$ exceeding the tabular value of the Fisher's $F$-test testified to the adequacy of the model.

### 4.2. Results of the Study of the Regularities of the Influence of Traffic Light Control on Traffic Flow Rate

In a similar way, the initial data on the process of changing the traffic flow rate under the influence of various characteristics of the traffic light control cycle were obtained (Table 7). The numerical values of saturation flows were determined by the calculation method based on the obtained measurements of the geometric characteristics of the intersection. The initial data on the geometric characteristics of the intersection and approaches to it was obtained by measuring the distances made to scale and plotted on the layout of the technical means of traffic organization at the intersection. The layout was provided by the Department of Road Infrastructure and Transport of the administration of the city of Tyumen in electronic form in the ".dwg" data file format.

**Table 7.** Traffic light control cycle characteristics and saturation flow values.

| Traffic Light Operation Period | Traffic Direction | Saturation Flow $M_s$, Cars/Hour | Permitting Signal Duration $t_0$, s | Traffic Light Cycle Duration $T_c$, s | Share of the Permitting Signal in the Cycle $\lambda$ |
|---|---|---|---|---|---|
| 07:00–08:00 | Left-turn | 1760 | 86 | 160 | 0.54 |
| | Forward | 1906 | 36 | 160 | 0.23 |
| 08:00–09:00 | Left-turn | 1760 | 76 | 155 | 0.49 |
| | Forward | 1906 | 41 | 155 | 0.26 |
| 09:00–15:00, 19:00–23:00 | Left-turn | 1760 | 53 | 140 | 0.38 |
| | Forward | 1906 | 49 | 140 | 0.35 |
| 15:00–19:00 | Left-turn | 1760 | 68 | 155 | 0.44 |
| | Forward | 1906 | 49 | 155 | 0.32 |

A graphical display of the model of the influence of the traffic light control cycle on the traffic flow rate (17) is shown in Figure 9.

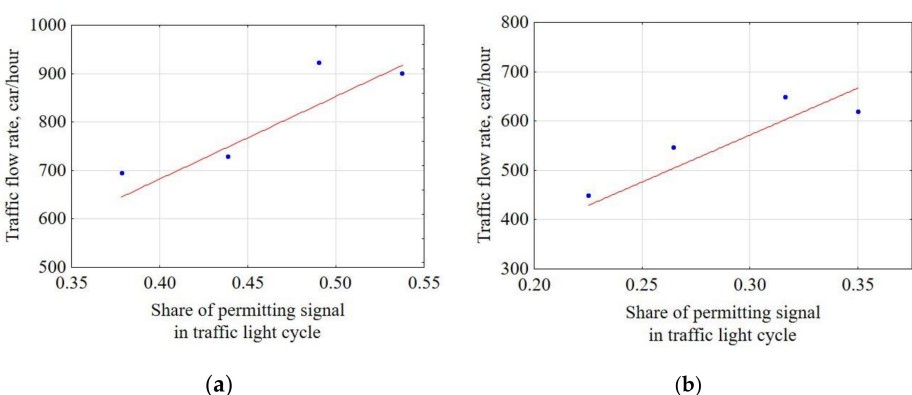

(**a**)　　　　　　　　　　　(**b**)

**Figure 9.** The influence of the traffic light control cycle on the traffic flow rate in the left-turn (**a**) and forward (**b**) directions of the traffic flow.

For the mathematical model (17), the numerical values of its parameters and statistical characteristics were determined (Table 8).

**Table 8.** Numerical values of the parameters and statistical characteristics of the model of the influence of the traffic light cycle on the traffic flow rate.

| Parameters and Statistical Characteristics of the Model | Left-Turn Direction of the Traffic Flow | Forward Direction of the Traffic Flow |
|---|---|---|
| Saturation flow $M_s$, cars/hour | 1760 | 1906 |
| Correlation coefficient $r$ | 0.86 | 0.81 |
| Student's $t$-test (tabular) | 2.92 | 2.92 |
| Student's $t$-test (calculated) | 3.38 | 3.1 |
| Determination coefficient $R^2$ | 0.74 | 0.71 |
| Fisher's variance ratio $F$ | 4.17 | 4.16 |
| Fisher's $F$-test | 4.11 | 4.11 |

The numerical values of the determination coefficients of the model of the influence of the traffic light control cycle on the traffic flow rate for the left-turn and forward directions of the traffic flow were 0.74 and 0.71, respectively, and the correlation coefficients were 0.86 and 0.84, respectively, which also confirmed the high strength of the relationship between the variables. The excess of the obtained calculated value of the Student's $t$-test compared to the tabular one confirmed the significance of the obtained correlation, and the Fisher's

variance ratio *F* exceeding the tabular value of the Fisher's *F*-test testified to the adequacy of the proposed model.

### 4.3. Results of the Study of the Regularities of the Combined Influence of Traffic Flow Concentration in Time and Traffic Light Control on Traffic Flow Rate

Table 9 presents the parameters and statistical characteristics determined for the mathematical model (18) describing the combined effect of the lane occupancy and traffic light cycle.

**Table 9.** Parameters and statistical characteristics of the model of the combined influence of lane occupancy and traffic light control cycle on the traffic intensity of the traffic flow.

| Traffic Direction | Model Parameters | | Saturation Flow $M_H$ Cars/Hour | Optimal $\theta$, % | Correlation Coefficient R | Student's t-Test | Determination Coefficient $R^2$ | Average Approximation error, % | Fisher's Variance Ratio F |
|---|---|---|---|---|---|---|---|---|---|
| | $a_1$, 1/% | $b_1$, 1/% | | | | | | | |
| Left-turn | $342 \cdot \times 10^{-6}$ | $370 \cdot \times 10^{-4}$ | 1760 | 62 | 0.90 | 12.5 | 0.82 | 8.92 | 2.02 |
| Forward | $360 \cdot \times 10^{-6}$ | $383 \cdot \times 10^{-4}$ | 1906 | 48 | 0.86 | 9.68 | 0.74 | 11.83 | 1.71 |

The numerical values of the coefficients of determination and correlation indicated a high relationship between the variables. The excess of the obtained calculated value of the Student's *t*-test compared the tabular one confirmed the significance of the obtained correlation. The average approximation error was within acceptable limits, and the Fisher's variance ratio *F* exceeded the tabular value of the Fisher's *F*-test, which together testified to the adequacy of the proposed model.

The surface shown in Figure 10 is a graphic representation of the combined effect of lane occupancy and the share of the permitting signal in the traffic light cycle on the traffic flow rate.

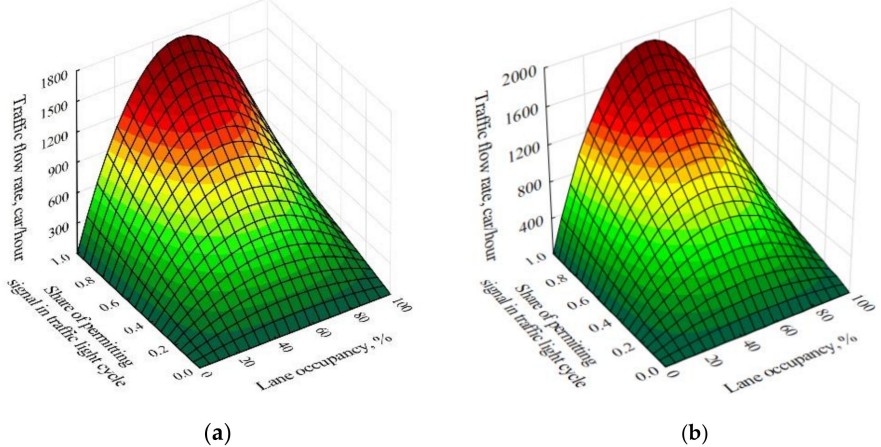

(**a**)          (**b**)

**Figure 10.** A combined influence of lane occupancy and traffic light control cycle on the traffic flow rate in left-turn (**a**) and forward (**b**) directions of the traffic flow.

### 4.4. Analysis of the Results Obtained

The obtained study results showed that traffic flow concentration in time, the measure of which is lane occupancy, can vary in the range of values from 0 to 100%. The minimum value of occupancy indicates the complete absence of vehicles in the lane, and the maximum value indicates that the movement of vehicles is completely stopped due to various reasons: vehicle breakdown, traffic accident, etc.

It is also possible to assert that there is an optimal value of lane occupancy at which the maximum possible value of the traffic flow rate is achieved, comparable to the value of the saturation flow. Therefore, further management of traffic flows at controlled city

intersections should be associated with the optimization of lane occupancy in all directions of traffic flows.

In turn, as the values of lane occupancy increase from zero to the optimal one, the traffic flow rate increases from zero to the maximum. In this case, there is a surplus in the capacity of the traffic lane under consideration, which can be used as a resource for adjusting the traffic light control cycle.

After traffic flow concentration in time reaches the optimal value, traffic flow rate decreases from maximum to zero, followed by an increase in lane occupancy to a critical 100% value. In this case, there is a deficit in the capacity of the traffic lane under consideration, which first indicates first the risk of traffic congestion, then the transition of the traffic flow to a congestion state, and ultimately the complete cessation of traffic on the investigated section of the road network.

The results of experimental studies have also shown that the optimal value of lane occupancy changes depending on the directions of traffic flow along the lanes. The authors believe that the change in the optimal value of lane occupancy depending on traffic directions is due to the fact that for the purpose of traffic safety in turning directions, drivers are forced to reduce the speed of movement. As a result, vehicles spend more time in the control zone of the detector than when driving in a forward direction. This phenomenon also determines the formation of the saturation flow value in turning directions, which was taken into account by the authors in this work when studying the influence of the traffic light control cycle on the maximum possible value of the traffic flow rate. In turn, the change in the share of the permitting signal in the traffic light cycle directly proportionally affects the maximum possible value of the traffic flow rate. An increase and decrease in the share of the permitting signal in the traffic light cycle additionally increases or limits the maximum possible value of the traffic flow rate, respectively, which was also experimentally confirmed in this work.

## 5. Discussion

At present, for many developed countries of the world, the most significant problem in the field of urban traffic organization is the formation of traffic congestion on the urban road network [1]. Previous studies note that this phenomenon is formed due to the high level of motorization [1,4,5].

Based on the results of the analysis of the available world expertise in solving this problem, three main approaches were identified: the road-building approach [1,8], the organizational and administrative approach [1,8–19], and the approach consisting of the use of intelligent transport systems [8,20–24]. According to the authors, the most promising one is the approach consisting of the use of intelligent transport systems to control traffic flows. It should be noted that in this matter, the authors of this work do not state it is the ultimate truth, and rightly emphasize that each of the conventionally formed approaches to solving the problems of traffic management has the right to exist, and can be used and be effective depending on the situation, available resources, and the goals and objectives of the researchers.

To select a key indicator of traffic management, the authors studied the process of traffic congestion on the city road network. It has been established that the formation of traffic jams is directly related to the process of changing the traffic flow rate [1,6,7]. This indicator was designated as a target for further research. The results of the analysis also showed that situations when the value of the traffic flow rate tends to zero arise not only in the event of a traffic jam, but also in the case of a complete absence of vehicles on the road network. To resolve this uncertainty, it is necessary to additionally take into account the data on the concentration of the traffic flow and compare them with the number of passing vehicles [26,27]. Based on previous studies, it was established that at present, for a number of reasons given by the authors as arguments, it is advisable to use the concentration of traffic flow in time, the measure of which is lane occupancy [7,27–33].

The results of the analysis of previous studies also showed that the presence of traffic lights at city intersections is certainly necessary to improve road safety and minimize the number of road accidents, but it significantly reduces the maximum possible traffic flow rate, which further exacerbates the situation and contributes to the formation of traffic congestion [34–37].

Unfortunately, the combined effect of the concentration of traffic flow in time and the means of traffic light control on the traffic flow rate in cities has not been fully studied, which served as the basis for setting the goal of this work.

To describe the regularity of the process of changing the traffic flow rate under the influence of lane occupancy, a one-factor quadratic mathematical model was developed based on a macroscopic hydrodynamic model of traffic flow [26,44]. Despite the existing criticism regarding the inoperability of the selected model under low-concentration conditions, the authors of this work adhere to the position that the indicated drawback of the selected model is associated with the density indicator, which has been used until now as the main measure of traffic flow concentration. The discrepancy between theoretical positions and experimental data can be associated with the problem of measuring density and the impossibility of obtaining correct information under normal conditions, which was noted by the authors based on previous studies. At the same time, the question of the relationship or its absence between the indicators of the concentration of the traffic flow in time and space currently also remains unanswered. On the one hand, there are previous studies that indicate the possibility of a relationship between these indicators [27]. On the other hand, it has not yet been possible to experimentally confirm the existence of this relationship in full [25,26,30]. Within the framework of this work, the authors carried out additional studies that also confirmed at least the existence of a theoretical relationship between traffic density and lane occupancy. Subsequently, the results obtained made it possible to transform the initial hydrodynamic model, taking into account the effect of lane occupancy on the traffic flow rate, and also to form a working hypothesis that subsequently was confirmed experimentally.

## 6. Conclusions

Theoretical and experimental studies performed by the authors of this paper in accordance with the designated methodology on the example of one of the most significant street intersections in the city of Tyumen in the Russian Federation confirmed the existence of regularities in the process of changing traffic flow rate under the influence of traffic concentration and traffic light control. The results also showed that this process is characterized by the presence of an optimum; i.e., the optimal lane occupancy value at which the maximum traffic flow rate is achieved. A deviation from the optimum indicates the irrational use of the road network resource, which in turn indicates the need to optimize road traffic by redistributing traffic flows. Experimental studies also confirmed the adequacy of the developed mathematical model of the process under study, and made it possible to determine its parameters depending on the directions of traffic flow. The analysis of the experimental results showed that depending on the direction of movement of vehicles, the optimal value of the lane occupancy also changes. The authors believe that this phenomenon is justified by a decrease in the speed of the traffic flow in order to safely perform the turn. Thus, when driving in a cornering direction, the maximum value of the traffic flow rate is formed with a larger value of lane occupancy.

The obtained results of the study can be directly used to improve the algorithms for the operation of controlled intersections as part of urban automated traffic control systems. The developed mathematical model will make it possible to predict the maximum value of the traffic flow rate at the exit of the controlled intersection, taking into account not only the current operating mode of traffic lights, but also the influence of lane occupancy. In the future, this will make it possible to determine the deficit or surplus of the time required to meet the actual transport demand at the controlled intersection, and thereby make a more accurate adjustment of the traffic light cycle.

Therefore, further promising areas of research will be the development of a practical methodology aimed at increasing the efficiency of traffic management, taking into account the traffic flow concentration in time, as well as clarifying the regularities of the influence of lane occupancy under different conditions of the road surface and weather conditions.

**Author Contributions:** Conceptualization, V.M.; methodology, S.I.; validation, V.M. and S.I.; V.M. analyzed the state of the issue to solve the indicated problem; developed the hypothesis and theoretical and experimental parts of the study; analyzed the results of the study; and wrote the text of the paper. All authors have read and agreed to the published version of the manuscript.

**Funding:** This research received no external funding.

**Acknowledgments:** The article was prepared as part of the implementation of a state assignment in the field of science for scientific projects carried out by teams of researchers in scientific laboratories of higher educational institutions subordinate to the Russian Ministry of Education and Science for the project: "New patterns and solutions for the functioning of urban transport systems in the paradigm 'Transition from owning a personal car to mobility as a service' " (No. 0825-2020-0014, 2020–2022).

**Conflicts of Interest:** The authors declare no conflict of interest.

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
