# Peer review of "Formation of the Traffic Flow Rate under the Influence of Traffic Flow Concentration in Time at Controlled Intersections in Tyumen, Russian Federation"

_sustainability, doi:10.3390/su13158324_

Round 1

Reviewer 1 Report

Dear Editor,

Dear authors,

Thank you very much for the opportunity to read the manuscript “Formation of the traffic flow rate under the influence of traffic flow concentration in time at controlled intersections in Tyumen, the Russian Federation”.

The objective of this study “is to establish the regularities of the influence of the concentration of traffic flow in time on the traffic flow rate at controlled city intersections” based on a system approach, using a diversity of tools and software to collect and treat the data, and having the city of Tyumen, Russian Federation, as case study.

In general, the manuscript follows the usual structure of a research article and presents work done based on primary data. However, some major concerns were identified: some portions of the text are located in inadequate sections (especially the information present in the results sections, as part of it is clearly methods and other part have a theoretical basis). Considering the objective of the study “to establish the regularities of the influence of the concentration of traffic flow in time on the traffic flow rate at controlled city intersections” and considering the scarce references to the relevance of the subject to sustainability, this paper presents a more suitable scope for more specialized journal and more specialized audience on the subject, for example "Future Transport" by MDPI.

Beyond that, I point out some small notes to the text following the structure presented.

TITLE

  • The title and the keywords are very similar. In my opinion, authors could amplify the title, for example, referencing the relevance of the subject for transport planning in order do amplify the readers of the paper;

Suggestion: “Traffic flow rate based on intelligent transport systems at controlled intersections in Tyumen, the Russian Federation

ABOUT THE KEYWORDS

  • Despite the several references to “traffic congestion” within the text, being a keywork, the authors should provide their understanding of the concept / a definition, perhaps as the second sentence of the introduction;

ABSTRACT

  • The theoretical framework have a disproportionate dimension compared with the results. A better balance is suggested, reducing the theoretical part and increasing the reference to results and conclusions.

INTRODUCTION

  • Lines 64-65 – it is not clear the relation of COVID19 with the following sentence – “show that the creation of an effective traffic management system is impossible without the use of modern intelligent technologies.”. This should be clarified or the reference should be removed;
  • line 68 – “USA, Japan, Europe, Russia, and other developed countries of the world”. As it is, seems thar Europe is a country;

MATERIALS AND METHODS

  • lines 107-108 – Just a curiosity. In Tyumen there are three rush peaks (morning, afternoon, and evening)? Traditionally there is only two;
  • as we are talking about routes, the description of the 1st paragraph would be stronger if the authors provides a map with the route and the urban context;
  • Several paragraphs of the section 3. Results could be better positioned in this section;

RESULTS

3.1. Analysis of the state of the issue

- lines 136-140 – please, include the source;

- line 146 – “every second resident of the city has a car” is not very clear the meaning (perhaps every two residents of the city have a car);

- fig. 1 – according the represented data, a column graph will fit better; it is possible to erase “calendar” of x-axis label;

- fig. 2 – please, confirm the values as the sum is not 100%

- lines 168-178 – this paragraph could be imported to the introduction as this do not represent any result.

- lines 179-214 – again, this is not any result. The authors should consider creating a new section after the introduction including this matter that obviously is relevant but is not a result. This will, after, justify the methodology options;

3.2. Theoretical research

- lines 218-351 -  again, this is not any result. The authors should consider creating a new section after the introduction

- Lines – 445 – 473 – this is clearly methodology (should be included in the methods section)

- in general, the results should be deeply analysed and explained.

DISCUSSION

  • There is no references to other works to justify the discussion.

As suggested by the journal: Authors should discuss the results and how they can be interpreted in perspective of previous studies and of the working hypotheses. The findings and their implications should be discussed in the broadest context possible and limitations of the work highlighted. 

CONCLUSION

  • In the conclusion, results should not be described, but present something more reflexive about the subject and the contribution of the study in the subject.

Author Response

Dear Sir or Madam,

Thank you for a very detailed and thorough analysis of our scientific article! All the comments you sent have been reviewed and mostly corrected. We are sending you a corrected scientific article and a full description of the corrected comments. We ask you to be understanding, since the recommendations and comments of other reviewers were taken into account when correcting the article.

Your faithfully,
Viacheslav Morozov

1 TITLE.
The title and the keywords are very similar. In my opinion, authors could amplify the title, for example, referencing the relevance of the subject for transport planning in order do amplify the readers of the paper;
Suggestion: “Traffic flow rate based on intelligent transport systems at controlled intersections in Tyumen, the Russian Federation
TITLE.
Of course, our scientific article may have different titles. In our opinion, this scientific article is devoted to the regularities for the identification of which a mathematical model was developed and experimentally tested.This is fully reflected in the title we have proposed. Therefore, if your comment is of a recommendatory and not very strict nature, we would prefer to leave the title unchanged
2 ABOUT THE KEYWORDS.
Despite the several references to “traffic congestion” within the text, being a keywork, the authors should provide their understanding of the concept / a definition, perhaps as the second sentence of the introduction;

ABOUT THE KEYWORDS.

The term "traffic congestion" is understood by the authors as a phenomenon that is formed as a result of "a consequence of a decrease in the traffic flow rate on the section of the road network serving traffic flows in relation to the section of the road network that forms traffic flows". This definition is given in the third paragraph of the introduction.

3 ABSTRACT
The theoretical framework have a disproportionate dimension compared with the results. A better balance is suggested, reducing the theoretical part and increasing the reference to results and conclusions.
ABSTRACT
The abstract has been corrected in accordance with your recommendations
4 INTRODUCTION
1) Lines 64-65 – it is not clear the relation of COVID19 with the following sentence – “show that the creation of an effective traffic management system is impossible without the use of modern intelligent technologies.”. This should be clarified or the reference should be removed;
2) line 68 – “USA, Japan, Europe, Russia, and other developed countries of the world”. As it is, seems thar Europe is a country;
INTRODUCTION
1) the reference has been removed
2) the remark has been corrected
5 MATERIALS AND METHODS
1) lines 107-108 – Just a curiosity. In Tyumen there are three rush peaks (morning, afternoon, and evening)? Traditionally there is only two;
2) as we are talking about routes, the description of the 1st paragraph would be stronger if the authors provides a map with the route and the urban context;
3) Several paragraphs of the section 3. Results could be better positioned in this section;
MATERIALS AND METHODS
1) The authors believe that at present it is possible to designate the afternoon  peak, which, depending on the location of intersections in the city, is generally formed in the period from 12 p.m. to 2 p.m. local time. Most likely, this is due to the time of the lunch break in many urban enterprises and the end of the first (morning) shift in schools. This observation is confirmed by data on the traffic flow rates for at least the last 4 years.
2) The description of exits on the territory of the city is given as an argument when choosing the most significant regulated intersection for further research. Therefore, in our opinion, this remark is of a recommendatory nature.
3) Your offer has been taken into account in full. The lines of the article that you indicated in the following remarks have been moved to this section
6 RESULTS

3.1. Analysis of the state of the issue
1) - lines 136-140 – please, include the source;
2) - line 146 – “every second resident of the city has a car” is not very clear the meaning (perhaps every two residents of the city have a car);
3) - fig. 1 – according the represented data, a column graph will fit better; it is possible to erase “calendar” of x-axis label;
4) - fig. 2 – please, confirm the values as the sum is not 100%
5) - lines 168-178 – this paragraph could be imported to the introduction as this do not represent any result.
6) - lines 179-214 – again, this is not any result. The authors should consider creating a new section after the introduction including this matter that obviously is relevant but is not a result. This will, after, justify the methodology options;

3.2. Theoretical research
7) - lines 218-351 -  again, this is not any result. The authors should consider creating a new section after the introduction
8) - Lines – 445 – 473 – this is clearly methodology (should be included in the methods section)
9)- in general, the results should be deeply analysed and explained
RESULTS

3.1 Analysis of the state of the issue
1) The source was included by the authors in the names of the diagrams, and it was also included by the authors in the previous paragraph of the work. Nevertheless, the source was also included in the paragraph you specified
2) The authors meant that on average 50 percent of the residents of the city of Tyumen have a vehicle. The remark has been corrected
3) Your offer has been taken into account in full
4) This remark is caused by the error of rounding numbers to their integer values. The remark has been corrected
5) Your offer has been taken into account in full. In the article, after the chapter "Introduction", a chapter "Analysis of the state of the issue" was introduced"
6) Your offer has been taken into account in full. In the article, after the chapter "Introduction", a chapter "Analysis of the state of the issue" was introduced"

3.2. Theoretical research
7) In accordance with your comment, as well as the comments and suggestions of other reviewers, this lines have been  included in section 2 " Methods and Methodology"
8) Your offer has been taken into account in full. This lines have been  included in section 2 " Methods and Methodology"
9) In accordance with your recommendations, an in-depth analysis and explanation of the results obtained is presented at the end of the "Results" section.
7 DISCUSSION
1) There is no references to other works to justify the discussion.
2) As suggested by the journal: Authors should discuss the results and how they can be interpreted in perspective of previous studies and of the working hypotheses. The findings and their implications should be discussed in the broadest context possible and limitations of the work highlighted.  
DISCUSSION
1)Your remark was taken into account in full
2) The discussion section has been updated taking into account your comments and the requirements of the journal
8 CONCLUSION
In the conclusion, results should not be described, but present something more reflexive about the subject and the contribution of the study in the subject.
CONCLUSION
In accordance with your remark, the unnecessary description of the research results was removed in the "Conclusions" section and information regarding the the contribution of the study in the subject.

Reviewer 2 Report

The authors dealt with the issue of the impact of traffic flow concentration in time on the formation of the traffic flow rate at controlled intersections. In a dense transport network, even minor disturbances of a single traffic stream may lead to deterioration of traffic conditions and cause the appearance of the congestion effect. This, in turn, increases delay, travel time, and additional traffic costs. Therefore, effective traffic management is an important tool to help minimize disruptions and prevent the creation and spread of traffic congestion. Thus, Therefore, I consider the research carried out by the authors to be justified and very significant.

However, I have the following comments on the manuscript.

- In my opinion, the title is not worded properly. I suggest: „Influence of traffic flow concentration in time on the formation of the traffic flow rate at controlled intersections in Tyumen, the Russian Federation”.

- The introduction should contain a brief description of the structure of the article.

- In my opinion, Chapter 2 is too little developed. The description of the methodology here is not very precise. The Authors present the adopted approach, but later in the text. Some of the content that, in my opinion, should be included in Chapter 2, has been placed in the next chapter (Chapter 3 Results), which in turn should contain (as the name suggests) the results of the research, and not the description of the applied methodology. As a result, the whole description is too chaotic and requires proper arrangement.

- Subsection 3.1. (The analysis of the state of the issue) should not be included in the part describing the research results, but much earlier (in the introduction or as a separate chapter).

- I propose to separate the chapter describing the research area (or as a subsection in chapter 2). Currently, the description of the research area is mixed with the methodology (page 5). I suggest moving the text from lines 168 - 214 to the part describing the research methodology. This comment also applies to subsection 3.2 (up to line 447).

Concluding, I believe that the proposed approach has great potential and possibilities of application. The number of publications cited is also impressive. To sum up, the results can be very interesting, but due to the lack of order, it is difficult to assess their substantive value.

Author Response

Dear Sir or Madam,
Thank you for your comments and recommendations for correcting and improving our article! If we understood correctly, in general, your assessment of our article was positive. We are very pleased and thank you for such a high assessment of our article. We are sending you a corrected scientific article and a full description of the corrected comments. We ask you to be understanding, since the recommendations and comments of other reviewers were taken into account when correcting the article.

Your faithfully,
Viacheslav Morozov

1 - In my opinion, the title is not worded properly. I suggest: „Influence of traffic flow concentration in time on the formation of the traffic flow rate at controlled intersections in Tyumen, the Russian Federation”. Of course, our scientific article may have different titles. In our opinion, this scientific article is devoted to the regularities for the identification of which a mathematical model was developed and experimentally tested.This is fully reflected in the title we have proposed. Therefore, if your comment is of a recommendatory and not very strict nature, we would prefer to leave the title unchanged
2 - The introduction should contain a brief description of the structure of the article. The brief description of the structure of the article was introduced in the chapter "Intriduction"
3 - In my opinion, Chapter 2 is too little developed. The description of the methodology here is not very precise. The Authors present the adopted approach, but later in the text. Some of the content that, in my opinion, should be included in Chapter 2, has been placed in the next chapter (Chapter 3 Results), which in turn should contain (as the name suggests) the results of the research, and not the description of the applied methodology. As a result, the whole description is too chaotic and requires proper arrangement. In accordance with this and the others of your comments, as well as the comments of other reviewers, the structure of this scientific article has been revised. Currently, the names of the chapters fully correspond to their content.
4 - Subsection 3.1. (The analysis of the state of the issue) should not be included in the part describing the research results, but much earlier (in the introduction or as a separate chapter). Your offer has been taken into account in full. In the article, after the chapter "Introduction", a chapter "Analysis of the state of the issue" was introduced"
5 - I propose to separate the chapter describing the research area (or as a subsection in chapter 2). Currently, the description of the research area is mixed with the methodology (page 5). I suggest moving the text from lines 168 - 214 to the part describing the research methodology. This comment also applies to subsection 3.2 (up to line 447). In accordance with your previous comment, lines 168-214 have been included to a new chapter.
Also, in accordance with the comments and suggestions of other reviewers, lines 218-473 were fully included in section 2 " Methods and Methodology"

Reviewer 3 Report

In this paper, the authors test several methods for traffic flow modeling at controlled city intersections. Although the topic has relevance, the current version of the paper needs severe improvement to make the paper suitable for publication.  In particular:

  • The main goals of the paper are not sufficiently clear. At the moment, several methods are tested and discussed, but the paper looks too much like a mix of many different ideas.
  • The structure of the paper needs to be reconsidered. The authors provide a very short “Materials and Methods” section and then continue with a very long “Results” section, which contains much information that one usually considers as part of the “Materials and Methods” section. Besides, the authors should introduce different subsections to ease the reading of the paper.
  • The authors insist both in the abstract and in the “Introduction” about the transition from spatial to temporal flow concentration. I think that both the temporal and the spatial components are important, so I cannot understand the authors’ statements in this regard.

Author Response

We thank you for your honest and specific feedback on our scientific article. We recognize and understand that the first version of our scientific article could contain comments of various nature. Thanks, among other things, to your comments and recommendations, we were able to edit and improve our scientific article. We ask you to be understanding, since the recommendations and comments of other reviewers were taken into account when correcting the publication.

Your faithfully,
Viacheslav Morozov

1 The structure of the paper needs to be reconsidered. The authors provide a very short “Materials and Methods” section and then continue with a very long “Results” section, which contains much information that one usually considers as part of the “Materials and Methods” section. Besides, the authors should introduce different subsections to ease the reading of the paper. The scientific publication has been significantly corrected.
Some of the materials from the "Results" section have been moved to the "Materials and Methods" section. A new section has been introduced, subsections have been introduced.
2 The authors insist both in the abstract and in the “Introduction” about the transition from spatial to temporal flow concentration. I think that both the temporal and the spatial components are important, so I cannot understand the authors’ statements in this regard. As it was indicated in our scientific publication, in addition to the already established approach to using the spatial traffic flow concentration, the approach to using the temporal traffic flow concentration is also becoming widespread in the world. We fully agree with you that depending on the goals, tasks and resources, a researcher can use one or another parameter. In our opinion, it is more rational to resort to the temporal traffic flow concentration, and we argued with the information provided in Table 1 and further to the text.

Round 2

Reviewer 1 Report

Dear Authors,

Thank you for revising your manuscript considering our indications.

In the new version of the manuscript, all major corrections were made, and no changes were explained by the authors. The article has a clearer and more organized structure.

Perhaps the only point that I suggest a small improvement is the conclusion, as it starts right away with the future potential of the tool, not reflecting on the conclusions of the study itself (which could be presented in an initial paragraph of the conclusion).

Best regards.

Author Response

Dear Sir or Madam,
Thank you for evaluating our new version of the scientific publication.
Your recommendation has been fully taken into account.
A paragraph has been added to the "Conclusion" section, briefly describing the main results of the study.
Your faithfully,
Viacheslav Morozov

Reviewer 2 Report

Dear Authors,

Thank you for the comprehensive replies to my comments and for correcting the manuscript as suggested. Currently, the content is much more structured. In this version, the research problem and the obtained results have been presented more clearly.

I have no substantive objections to the article.

Best regards

Author Response

Dear Sir or Madam,

Thank you for your next review and such a high assessment of our scientific article.

In general, the other reviewers also positively characterized our new version of the scientific article. However, one of the reviewers left minor recommendations for improving our article. We have taken these recommendations into account. In this regard, we also send you a new version of the scientific article for review..

Your faithfully,
Viacheslav Morozov

Reviewer 3 Report

I have no further comments.

Author Response

Dear Sir or Madam,

Thank you for evaluating the new version of our scientific article. Despite the fact that your new review does not contain comments, one of the reviewers outlined minor recommendations for improving our scientific article. We have taken these recommendations into account and consider it necessary to send you a new version of our scientific article.

Your faithfully,
Viacheslav Morozov
